# Clinical Pattern of Preoperative Positron Emission Tomography/Computed Tomography (PET/CT) Can Predict the Aggressive Behavior of Resected Solid Pseudopapillary Neoplasm of the Pancreas

**DOI:** 10.3390/cancers13092119

**Published:** 2021-04-27

**Authors:** Ji-Su Kim, Emmanuel II-Uy Hao, Seoung-Yoon Rho, Ho-Kyoung Hwang, Woo-Jung Lee, Dong-Sub Yoon, Chang-Moo Kang

**Affiliations:** 1Department of Hepatobiliary and Pancreatic Surgery, Yonsei University College of Medicine, Seoul 03722, Korea; kgc2608@yuhs.ac (J.-S.K.); DRHHK@yuhs.ac (H.-K.H.); WJLEE@yuhs.ac (W.-J.L.); YDS6110@yuhs.ac (D.-S.Y.); 2Pancreatobiliary Cancer Clinic, Severance Hospital, Yonsei University College of Medicine, Seoul 03722, Korea; 3Department of Surgery, Philippine General Hospital, University of the Philippines Diliman, Quezon City 1100, Philippines; euhao@up.edu.ph; 4Department of Surgery, Yongin Severance Hospital, Yongin-si 17046, Korea; FORSH7@yuhs.ac

**Keywords:** aggressive tumor, pancreas, pancreatic neoplasm, solid pseudopapillary tumor, aggressive pancreatic cancer

## Abstract

**Simple Summary:**

Predicting the aggressiveness of solid pseudopapillary neoplasms (SPNs) remains a worthwhile goal. The present study aimed to identify perioperative factors that can predict patients who will develop clinically aggressive SPN. A total of 98 patients diagnosed with SPNs were analyzed retrospectively. We found that age (≥40 years; *p* = 0.039), symptomatic presentation (*p* = 0.001), tumor size (>10 cm; *p* < 0.001), positron emission tomography/computed tomography (PET/CT) classification (Type III; *p* < 0.001), and lymphovascular invasion (*p* = 0.003) were significantly correlated with aggressive behavior of SPNs. Among these, age ≥40 years, PET/CT Type III configuration, and lymphovascular invasion were independent factors associated with an aggressive SPN. This information can help clinicians develop individualized management and surveillance plans to manage patients more competently.

**Abstract:**

Predicting the aggressiveness of solid pseudopapillary neoplasms (SPNs) remains an important goal. The present study aimed to identify perioperative factors that can predict patients who will develop clinically aggressive SPN. Records of individuals with pathologically confirmed SPN from 2006 to 2017 were obtained from the patient registry database of Yonsei University, Severance Hospital. For this study, aggressive behavior was defined as SPN that had recurred, metastasized, or involved adjacent organs. A total of 98 patients diagnosed with SPNs were analyzed retrospectively. Of these, 10 were reported to have SPNs with aggressive characteristics. We found that age (≥40 years; *p* = 0.039), symptomatic presentation (*p* = 0.001), tumor size (>10 cm; *p* < 0.001), positron emission tomography/computed tomography (PET/CT) classification (*p* < 0.001), and lymphovascular invasion (*p* = 0.003) were significantly correlated with aggressive behavior of SPNs. Multivariate analysis showed that PET/CT configuration (*p* = 0.002) (exp(β)111.353 (95% confidence interval (CI): 5.960–2081), age ≥40 years (*p* = 0.015) (exp(β) 23.242 (95% CI: 1.854–291.4)), and lymphovascular invasion (*p* = 0.021) (exp(β) 22.511 (95% CI: 1.595–317.6)) were the only independent factors associated with aggressive SPN. Our data suggest that age ≥40 years, PET/CT Type III configuration, and lymphovascular invasion are independent factors associated with aggressive SPN. This information can help clinicians develop individualized management and surveillance plans to manage patients more effectively.

## 1. Introduction

Solid pseudopapillary neoplasm (SPN) of the pancreas is a low-grade malignant neoplasm that accounts for only 1–2% of all pancreatic neoplasms [1,2,3,4]. SPN of the pancreas is usually non-aggressive, but up to 10–15% of neoplasms have been reported to be aggressive [5]. Describing and characterizing the natural course of SPNs has been difficult owing to the rarity of this disease, as well as the inconsistent correlation between the pathological characteristics and clinical course.

Several studies have attempted to identify factors that could predict the aggressive behavior of SPNs, including age [6], sex [7], neutrophil-to-lymphocyte ratio [8], tumor size [5,9,10,11], resection margin [6], Ki-67 index [1,12,13,14], pathological variables, such as lymphovascular and perineural invasion [1,12,15], and radiologic findings on magnetic resonance imaging (MRI) and positron emission tomography/computed tomography (PET/CT) [16,17,18,19,20,21]. Meanwhile, several other studies have reported that there are no clinicopathologic factors that can predict tumor behavior [22,23,24,25,26].

Predicting the aggressiveness of SPNs remains a worthwhile goal. Identifying patients with aggressive SPNs can help surgeons create individualized clinical management and surveillance strategies [27,28]. This study aimed to identify perioperative factors that can predict patients who will develop clinically aggressive SPN.

## 2. Materials and Methods

### 2.1. Definition of Terms

In this study, aggressive behavior was defined as SPN that either locally invaded adjacent structures, developed recurrence, or had systemic metastasis, whether at initial diagnosis or at a later course.

### 2.2. Data Collection

Patient records from 2006 to 2017 mentioning SPN were obtained from the patient registry database of Yonsei University Severance Hospital. These records were then reviewed in search of a pathologically confirmed diagnosis of SPN of the pancreas. Perioperative data, including age, sex, body mass index, symptoms at presentation, operation date, and type of operation were collected and encoded into a spreadsheet. Preoperative ancillary data, such as imaging studies and PET/CT scan results, were also included, specifically, the fluorodeoxyglucose (FDG) uptake pattern on PET/CT, as described by Kang et al. [17]. There are five types of PET/CT uptake classifications: Type I (hot FDG uptake in the entire tumor portion), Type II (focal defect), Type III (multiple and geographic uptake), Type IV (focal uptake), and Type V (total defective type). Postoperative results, such as the complication rate, length of hospital stay, and pathologic reports, were also tabulated into the spreadsheet. All cases were reviewed by immunohistochemical staining, including beta-catenin and Ki-67. Patients who had undergone definitive disease-related surgeries at other institutions, as well as those who had a preoperative diagnosis of SPN but were yet to undergo definitive surgery, were excluded from this study.

### 2.3. Statistical Analysis of Baseline Characteristics

International Business Machines Corporation SPSS Statistics Version 23 (SPSS, Inc., Chicago, IL, USA) was used for all statistical analyses. Continuous variables are represented as the means ± standard deviations or medians with ranges, while categorical variables are represented as percentages or frequencies. Continuous variables were compared using Student’s *t*-test, while categorical variables were compared using Fisher’s exact test or the chi-squared test. A binary logistic regression analysis was used to determine whether the factors affecting aggressive behavior were independent of each other. A statistically meaningful value for predicting aggressive behavior was set as the cutoff value. Statistical significance was set at *p* < 0.05.

## 3. Results

### 3.1. General Patient Data

From 2006 to 2017, 98 patients had pathologically confirmed pancreatic SPN. The female to male ratio was 83:15 (or 5.53:1), and the average age of the participants was 36.03 years. Table 1 shows a summary of patient information.

### 3.2. Preoperative Evaluation

All patients underwent preoperative imaging (CT or MRI) to determine the neoplasm characteristics. However, only 86 patients (87.8%) underwent PET/CT. Table 2 summarizes the preoperative PET/CT findings.

### 3.3. Surgical Outcomes

Complete resection was achieved in all but one patient. Moreover, pathologic reports of all resected neoplasms indicated that the tumor margins were negative. Microscopic pathology was noted in 24 patients (24.5%). These patients either had capsular invasion, lymphovascular invasion, perineural invasion, or a combination of these pathologic findings. Cellular proliferation, as assessed using the Ki-67 index, was observed in 64 patients. Other immunohistochemical staining results were also recorded; however, pathological reports for these are heterogeneous. Twenty-nine patients developed complications, but most were managed conservatively. A total of six disease-specific deaths were reported, of which four were related to aggressive type SPNs. Table 3 summarizes the surgical outcomes of the patients in the study, and Figure 1 shows the disease-specific survival plots for patients with aggressive SPN versus those with non-aggressive SPN.

### 3.4. Patients with Aggressive Neoplasm

Ten patients (seven women, three men) had aggressive SPN; of these, six had systemic recurrences, while the remaining four had locally infiltrative neoplasms affecting adjacent organs, confirmed histologically. Of the six patients, one received conservative treatment and five received chemotherapy. Their average age was 45.5 years, with two patients, aged 10 and 12 years, managed by pediatric surgeons. Table 4 describes the profiles of the patients with aggressive SPNs. Further details are in the Appendix A.

### 3.5. Factors Affecting Aggressive Behavior

We found that age (≥40 years; *p* = 0.039), symptomatic presentation (*p* = 0.001), tumor size (>10 cm; *p* < 0.001), PET/CT classification (Type III; *p* < 0.001), and lymphovascular invasion (*p* = 0.003) were significantly correlated with aggressive behavior of SPNs. Meanwhile, sex, tumor location, presence of complications, capsular invasion, perineural invasion, and Ki-67 index were found to have no statistically significant correlation with aggressive behavior of SPN. Table 5 summarizes the results of the statistical analyses. Multivariate analysis showed that Type III PET/CT configuration (*p* = 0.002), age ≥40 years (*p* = 0.015), and lymphovascular invasion (*p* = 0.021) were the only independent factors associated with aggressive SPN. As independent factors with a 96.5% successful predictive value for aggressiveness, Type III PET/CT configuration had an exp(β) of 111.353 (95% confidence intervals (CI): 5.960 and 2081), while age ≥40 years and lymphovascular invasion had exp(β) of 23.242 (95% CI: 1.854 and 291.4) and exp(β) of 22.511 (95% CI: 1.595 and 317.6), respectively.

## 4. Discussion

Advanced age, specifically ≥40 years, was an independent variable in predicting the aggressiveness of SPNs. Aging is associated with numerous physiological changes and adverse medical events. Advanced age amplifies the impact of certain conditions, such as hypertension, diabetes, and certain tumors. Some studies have suggested that cellular/physical degeneration and immune-related deterioration can be the cause of age-related disease conditions [30,31,32].

PET/CT uses the enhanced glucose metabolism of cancer cells as a basis for differential weighing and eventual detection of cancer cells [17,19,20]. An inherent characteristic of neoplasms is that they replicate at an unregulated rate, presumably at a rate faster than normal cells. We investigated the potential correlation between the clinical pattern of preoperative PET/CT and the biological behavior of resected SPN of the pancreas [17,33]. Notably, we presented neoplasms with varying uptake patterns, while having a relatively low Ki-67 index. The SPNs in our study presented all five types of uptake patterns, with an almost equal distribution between types I and IV (27.9%, 20.9%, 26.7%, and 20.9%), and Type V was the least frequent (3.5%). We also report that the Type III uptake pattern is an independent factor in predicting aggressive type SPN. The reason for this remains unclear, and requires further study. We recognize that other measurable parameters, such as the mean and maximum standard uptake values, metabolic tumor volume, and total lesion glycolysis, can provide information for the quantitative evaluation of neoplasms and their aggressiveness [19].

While neoplasm growth varies significantly, that is, some neoplasms grow faster than others, tumor size has always been associated with time. In general, the longer the neoplasm is left undetected, the larger it can potentially become. De Robertis et al. reported that large SPNs were more frequently located in the pancreatic body or tail (≥51 mm tumor location: head (17.6%), body (32.4%), tail (47.4%), *p* = 0.008) [34]. This study explained that this result is associated with a delayed diagnosis compared to that of pancreatic head tumors because of the potential for maximal tumor growth without symptoms of premature closure, such as jaundice and duodenal obstruction [34]. Thus, as the neoplasm enlarges, it can potentially affect adjacent structures, as well as metastasize to distant organs, as in the case of malignant lesions. This may rationalize our finding that the presence of lymphovascular invasion, often seen in large (≥10 cm) neoplasms, is a factor related to poor patient outcomes.

Similar to our findings, in a multicenter analysis in Korea, large tumors (>8 cm) were reported to be predictive of recurrence (exp(β) = 7.385, *p* = 0.018) [10]. Several studies have reported the importance of PET/CT as a predictor of aggressive behavior in SPNs [19,20]. Aisheng et al. reported that CT or MRI demonstrated morphological features of SPN, and FDG PET/CT reflected the histopathological composition of the tumors [20]. They explained that FDG uptake by SPN may be related to tumor cellularity, proliferative index, or histological malignancy [20].

Contrary to our results, some studies reported that young age and male sex were associated with SPN recurrence [6,7]. Sabine et al. reported that younger children had a high risk of recurrence (*p* = 0.03) [6]. Matthew et al. reported that male patients have an atypically aggressive biology of SPN [7]. Males had approximately twice the rates of metastases and invasive malignancy, and a threefold higher death rate than that of females (*p* = 0.036, *p* = 0.003, *p* = 0.002, respectively) [7]. In our study, there was no difference in the incidence of aggressive SPN in younger individuals under 18 years of age (*p* = 0.664), and aggressive SPN was identified as a risk factor for individuals who were 40 years of age or older (exp(β) = 23.242, *p* = 0.015). In addition, there was no statistically significant difference between males and females for aggressive SPN (*p* = 0.173).

Studies have shown that SPNs of the pancreas occur because of a somatic point mutation in exon 3 of CTNNB1, which encodes the β-catenin pathway [12,35]. This distinguishes it from other known pancreatic neoplasms. Meng et al. conducted whole exome sequencing in nine patients with SPN, and found that the CTNNB1 mutation potently collaborated with other gene variations [36]. Shmuel et al. reported that a panel of six miRNAs, including miR-184, miR-10a, miR-887, miR-217, miR200c, and miR-375, were significantly expressed in metastatic SPNs. These specific miRNAs have potential as predictive markers of aggressive behavior of SPN [37]. However, despite our knowledge of its genetic basis, the natural and clinical course of the disease remains unclear.

The results of this study presented clinical predictors, including PET/CT, to provide the basis for implementing PET/CT when treating patients with SPN in clinical practice. In patients with symptoms at diagnosis, large tumors (≥10 cm), and old age (≥40 years), PET/CT can be performed to check for aggressive behavior. The Type III uptake pattern of PET/CT can be considered as an indicator for active surgery and postoperative chemotherapy. To the contrary, patients with Type IV and V, mostly containing a defective background with minimal or null FDG uptake, could be carefully followed without surgery according to the patient’s individual conditions, such as co-morbidity, refusal of surgery, and active personal schedule [33]. These SPNs may show very indolent biological behavior or total necrosis [33]. Therefore, the results of this study are expected to help SPN treatment decisions.

This study has limitations related to its retrospective design and the small size of data. Moreover, there was a selection bias, since we only analyzed patients who underwent surgery. We recommend further studies on the potential of qualitative parameters in PET/CT scans as predictive factors for aggressiveness. We also recommend further research into the clinical course of SPN, along with its immunohistochemical and preoperative laboratory profiles, as well as the molecular basis of perilesional inflammation, to fully understand the pathophysiology of this neoplasm.

## 5. Conclusions

In conclusion, our data suggest that, in patients with SPN, age ≥40 years, symptomatic at presentation, tumor size ≥10 cm, and PET/CT scan configuration Type III are all predictive of aggressive neoplasm behavior. Lymphovascular invasion of the neoplasm, as seen in the pathological report, can also be a valuable predictor. Among these, age ≥40 years, PET/CT Type III configuration, and lymphovascular invasion are independent factors associated with an aggressive SPN. This information can help clinicians develop individualized management and surveillance plans to manage patients more competently.

Complete surgical resection remains the standard management for SPNs, but diligent surveillance may be warranted to detect recurrence or metastasis.

## Figures and Tables

**Figure 1 cancers-13-02119-f001:**
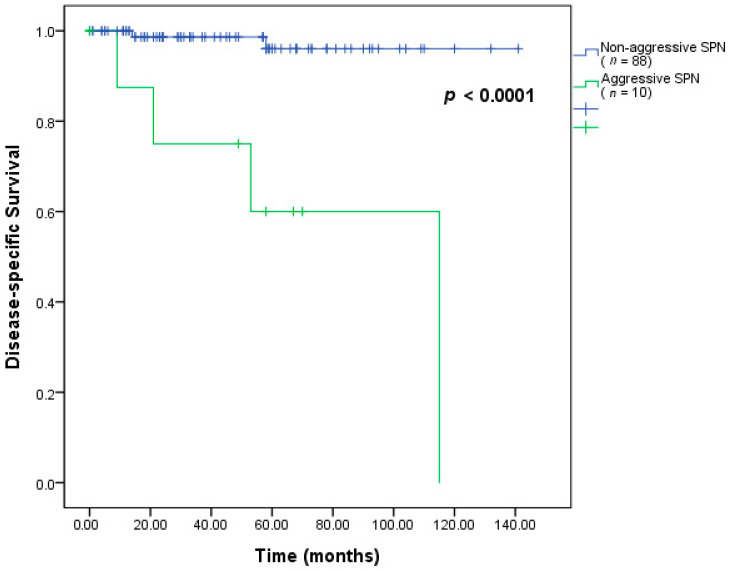
Disease-specific survival plot for aggressive versus non-aggressive SPN; SPN, solid pseudopapillary neoplasm.

**Table 1 cancers-13-02119-t001:** General patient data of all 98 patients.

Variables	No. of Patients (%)
Age	36.03 ± 15.6
Sex (F/M)	83 (84.7%)/15 (15.3%)
BMI	22.05 ± 3.3
Symptomatic (yes/no)	42 (42.9%)/56 (57.1%)
Abdominal pain	35 (83.3%)
Abdominal mass	9 (29.4%)
Tumor size (cm)	4.6 ± 2.8
Tumor location (proximal/distal)	29 (29.6%)/69 (70.4%)
Operation type (minimally invasive/open)	64 (65.3%)/34 (34.7%)
Operation	-
PPPD	21 (21.4%)
Central pancreatectomy	9 (9.2%)
Distal pancreatectomy	62 (63.3%)
Enucleation	5 (5.1%)
Biopsy	1 (1.0%)

Values are *n* (%), mean ± standard deviation. BMI, body mass index; PPPD, pylorus-preserving pancreaticoduodenectomy.

**Table 2 cancers-13-02119-t002:** Clinical pattern of preoperative PET/CT in all 86 patients.

PET/CT Scan Uptake Classification ^1^	Scheme	Description	No. of Patients (%)	Representative PET/CT Image
Type I	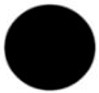	Completely hot uptake type, no defect area	24 (27.9%)	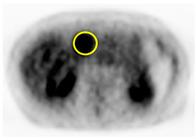
Type II	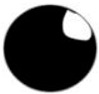	Focal defect area with almost type I background	18 (20.9%)	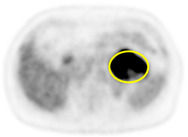
Type III	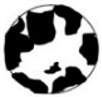	Multiple or geographic uptake with defective background	23 (26.7%)	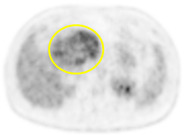
Type IV	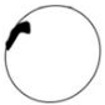	Focal uptake with mainly defective background	18 (20.9%)	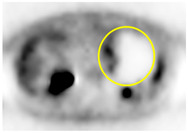
Type V	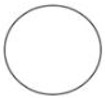	No uptake, completely defective background	3 (3.5%)	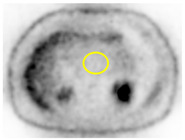

Values are *n* (%), mean ± standard deviation. PET/CT, positron emission tomography/computed tomography; ^1^ PET/CT scan uptake classification was described in a previous report [17].

**Table 3 cancers-13-02119-t003:** Surgical outcomes of all 98 patients.

Variables	No. of Patients (%)
Disease-specific deaths	6 (6.1%)
R0 resection	98 (100%)
Positive margins	0 (0%)
Microscopic pathology	-
Capsular invasion	24 (24.5%)
Lymphovascular invasion	35 (83.3%)
Perineural invasion	9 (29.4%)
Ki-67	4.6 ± 2.8
Complication ^1^	29 (29.6%)
Clavien–Dindo Class I	2 (2%)
Clavien–Dindo Class II	15 (15.3%)
Clavien–Dindo Class IIIa/IIIb	7 (7.1%)/3 (3.1%)
Clavien–Dindo Class IV/V	1 (1%)/1 (1%)

Values are *n* (%), mean ± standard deviation. ^1^ Complication was graded according to the Clavien–Dindo classification system [29].

**Table 4 cancers-13-02119-t004:** Profile of the patients with aggressive solid pseudopapillary tumors.

No.	Age (years)	Sex	Tumor Size (cm)	Symptom	Tumor Location	Type of Operation	Lymphovascular Invasion	Type of Aggressive Behavior	Treatment for Recurrence	Follow-Up Period (Status)
1	81	F	12	Abdominal mass	Distal	Distal pancreatectomy	None	Liver metastasis after five months	Conservative treatment	21 months (Death)
2	12	F	10.10	Abdominal mass	Distal	Distal pancreatectomy	None	Liver metastasis after 10 months	Chemotherapy	115 months (Death)
3	46	F	2.00	Incidental	Distal	Distal pancreatectomy	None	Peritoneal metastasis after 37 months	Chemotherapy	53 months (Death)
4	67	M	2.5	Abdominal pain	Distal	Distal pancreatectomy	None	Liver metastasis after 43 months	Chemotherapy	70 months (Survival)
5	45	F	11	Abdominal pain	Distal	Biopsy	None	Liver metastasis after two months	Chemotherapy	Nine months (Death)
6	46	F	4.3	Abdominal pain	Proximal	PPPD + metastasectomy	None	Liver metastasis on diagnosis; peritoneal metastasis after 41 months	Chemotherapy	58 months (Survival)
7	68	F	5	Abdominal pain	Proximal	PPPD + right hemicolectomy, PV resection	None	Invading hepatic flexure, portal vein	No recurrence	0 months (Death)
8	52	M	11	Abdominal pain	Proximal	PPPD + right hemicolectomy	None	Invading hepatic flexure	No recurrence	67 months (Survival)
9	10	F	6	Abdominal pain	Distal	Distal pancreatectomy + transverse colectomy	None	Invading transverse colon	No recurrence	49 months (Survival)
10	27	M	9.4	Abdominal mass	Proximal	PPPD + PV resection	None	Invading portal vein	No recurrence	39 months (Survival)

PPPD, pylorus-preserving pancreaticoduodenectomy, PV, portal vein.

**Table 5 cancers-13-02119-t005:** Characteristics of patients with aggressive and non-aggressive solid pseudopapillary tumor.

Variables	Aggressive (*n* = 10)	Non-Aggressive (*n* = 88)	*p*-Value	*p*-Value Exp(β) (95% CI)
Age	-	-	-	-
≥40 *	7 (70%)	32 (36.4%)	0.039	0.015 23.242 (1.854/291.4)
19–39	1 (10%)	43 (48.9%)	0.431	-
≤18	2 (20%)	13 (14.8%)	0.664	-
Sex (F/M)	7/3	76/12	0.173	-
BMI	21.7 ± 4.74	22.1 ± 3.10	0.804	-
Symptomatic (yes/no)	9/1	33/55	0.001	-
Location (proximal/distal)	4/6	25/63	0.447	-
Tumor size	-	-	-	-
≥10 cm	4 (40%)	2 (2.3%)	<0.001	-
≥5 cm	6 (60%)	26 (29.5%)	0.052	-
≥2 cm	9 (90%)	71 (80.7%)	0.471	-
PET configuration *	-	-	<0.001	0.002 111.353 (5.960/2081)
Type III	7 (77.8%)	16 (20.8%)	-	-
Non-Type III	2 (22.2%)	61 (79.2%)	-	-
Complication (yes/no)	4/6	26/62	0.497	-
Microscopic pathology	-	-	-	-
Margin	All negative	All negative	-	-
Capsular invasion	2 (20%)	16 (18.2%)	0.888	-
Lymphovascular invasion	3 (30%)	4 (4.5%)	0.003	0.021 22.511 (1.595/317.6)
Perineural invasion	3 (30%)	11 (12.5%)	0.134	-
Ki-67	3.42 ± 4.52	2.10 ± 2.52	0.266	-

Values are *n* (%), mean ± standard deviation; PET, positron emission tomography; BMI, body mass index; WBC, white blood cell; CI, confidence interval. * Independent variables for predicting the aggressiveness of SPNs.

## Data Availability

Data are available from the corresponding author upon request.

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
