# Peer review of "Clinical Pattern of Preoperative Positron Emission Tomography/Computed Tomography (PET/CT) Can Predict the Aggressive Behavior of Resected Solid Pseudopapillary Neoplasm of the Pancreas"

_cancers, 2021, doi:10.3390/cancers13092119_

Round 1

Reviewer 1 Report

Please correct at line 180 of in the discussion "Riccardo et al. reported …" into "De Robertis et al. reported … " 

Author Response

Author response to the reviewers' comments

Thanks again for your response and the reviewers' comments on our manuscript. We tried to faithfully revise our manuscript according to each question and concern raised by the reviewers, hoping that you will consider this manuscript for publication in Journal of Cancers

<Reviewer #1>

Please correct at line 180 of in the discussion "Riccardo et al. reported …" into "De Robertis et al. reported … " 

Comment 1: Please correct at line 180 of in the discussion "Riccardo et al. reported …" into "De Robertis et al. reported … " 

Response 1: As commented by the reviewer, we modified it correctly.

Revision 1:

(Line 180)

De Robertis et al. reported …

Reviewer 2 Report

The authors have made the requested revisions and strengthened the paper to the degree possible.  As is highlighted by the updated discussion, this manuscript does not add significantly to the existing literature and has limited impact. 

It appears that a peri-operative death was included in the survival analysis.  Because this could be related to a technical problem rather than disease biology, I suggest that patient mortality is excluding from survival analysis.

Author Response

Author response to the reviewers' comments

Thanks again for your response and the reviewers' comments on our manuscript. We tried to faithfully revise our manuscript according to each question and concern raised by the reviewers, hoping that you will consider this manuscript for publication in Journal of Cancers

<Reviewer #1>

The authors have made the requested revisions and strengthened the paper to the degree possible.  As is highlighted by the updated discussion, this manuscript does not add significantly to the existing literature and has limited impact. 

Comment 1 : It appears that a peri-operative death was included in the survival analysis.  Because this could be related to a technical problem rather than disease biology, I suggest that patient mortality is excluding from survival analysis.

Response 1 : As commented by reviewr, disease-specific survival have to exclude a peri-operative death. There was one patients with a peri-operative death. So, a total of 6 disease-specific death were reported, of which 4 were related to aggressive type SPNs. Survival analysis was again accurately performed, and all related contents (table, figure) were corrected.Thanks for your comment.

Revision 1 :

(Line 119,120)

A total of 6 disease-specific deaths were reported, of which 4 were related to aggressive type SPNs.

(Line 123)

Table 3. Surgical outcomes of all 98 patients.

Variables

No. of patients (%)

Disease-specific deaths

6 (6.1%)

(Line 127,128)

Figure 1. Disease-specific Survival plot for Aggressive versus Non-aggressive SPN

The figure is not shown. Please see the attachment

Round 2

Reviewer 2 Report

Comment addressed.

This manuscript is a resubmission of an earlier submission. The following is a list of the peer review reports and author responses from that submission.

Round 1

Reviewer 1 Report

Journal: Cancers

Aggressive Behavior of Solid Pseudopapillary Neoplasm of the 2 Pancreas: Can It Be Predicted?

The aim of this study was to predict the aggressiveness of Solid pseudopapillary neoplasm.

The article is dedicated on very important issue. The population is very numerous therefore of great interest for scientific community. The study is however focused on PET. This is not clear from the title so please introduce this aspect on the title or better add CT and MRI information.

The paper is not acceptable without PET images of different type.

Please insert the following recent paper.

Solid Pseudopapillary Neoplasms of the Pancreas: Clinicopathologic and Radiologic Features According to Size.

De Robertis R, Marchegiani G, Catania M, Ambrosetti MC, Capelli P, Salvia R, D'Onofrio M.AJR Am J Roentgenol. 2019 Nov;213(5):1073-1080. doi: 10.2214/AJR.18.20715. Epub 2019 Jul 16.PMID: 31310181

Author Response

Author response to the reviewers' comments

Thanks for your response and the reviewers' comments on our manuscript. We tried to faithfully revise our manuscript according to each question and concern raised by the reviewers, hoping that you will consider this manuscript for publication in Journal of Cancers

<Reviewer #1>

The aim of this study was to predict the aggressiveness of Solid pseudopapillary neoplasm.

The article is dedicated on very important issue. The population is very numerous therefore of great interest for scientific community.

Comment 1: The study is however focused on PET. This is not clear from the title so please introduce this aspect on the title or better add CT and MRI information.

Response 1: As suggested by the reviewer, we modified the title to include PET/CT.

Revision 1:

(Page 1)

Clinical pattern of preoperative positron emission tomography /computed tomography (PET/CT) can predict the aggressive behavior of resected solid pseudopapillary neoplasms of the pancreas

Comment 2: The paper is not acceptable without PET images of different type.

Response 2: As commented by the reviewer, PET image is very important in this study. We added the representative PET/CT images of different type in table 2.

Revision 2:

(Page 3,4)

Table 2. Clinical pattern of preoperative PET/CT in all 86 patients.

PET/CT scan uptake classification1

Scheme

Description

No. of patients (%)

Representative PET/CT image

Type I

Completely hot uptake type, no defect area

24 (27.9%)

Type II

Focal defect area with almost type I background

18 (20.9)

Type III

Multiple, or geographic uptake with defective background

23 (26.7%)

Type IV

Focal uptake with mainly defective background

18 (20.9%)

Type V

No uptake, completely defective background

3 (3.5%)

Values are n (%), mean ± standard deviation.

PET/CT, positron emission tomography/computed tomography;

1PET/CT scan uptake classification was described in a previous report [1].

  1. Kang, C.M.; Cho, A.; Kim, H.; Chung, Y.E.; Hwang, H.K.; Choi, S.H.; Lee, W.J. Clinical correlations with (18)fdg pet scan patterns in solid pseudopapillary tumors of the pancreas: Still a surgical enigma? Pancreatology : official journal of the International Association of Pancreatology (IAP) ... [et al.] 2014, 14, 515-523.

Comment 3: Please insert the following recent paper.

Solid Pseudopapillary Neoplasms of the Pancreas: Clinicopathologic and Radiologic Features According to Size.

De Robertis R, Marchegiani G, Catania M, Ambrosetti MC, Capelli P, Salvia R, D'Onofrio M.AJR Am J Roentgenol. 2019 Nov;213(5):1073-1080. doi: 10.2214/AJR.18.20715. Epub 2019 Jul 16.PMID: 31310181

Response 3: As suggested by the reviewer, we introduced the suggested paper as recent paper in discussion.

Revision 3:

(Page 7,8)

Riccardo et al. reported that large SPNs were more frequently located in the pancreatic body or tail (≥51 mm sized tumor location: head [17.6%], body [32.4%], tail [47.4%], p=0.008). This study explained that this result is associated with a delayed diagnosis compared to that of pancreatic head tumors because of the potential for maximal tumor growth without symptoms of premature closure, such as jaundice and duodenal obstruction [2].

  1. De Robertis, R.; Marchegiani, G.; Catania, M.; Ambrosetti, M.C.; Capelli, P.; Salvia, R.; D'Onofrio, M. Solid pseudopapillary neoplasms of the pancreas: Clinicopathologic and radiologic features according to size. AJR Am J Roentgenol 2019, 213, 1073-1080.

"Some pictures are not visible. Please see the attachment"

Reviewer 2 Report

This is a series of 98 patients with solid pseudopapillary neoplasm (SPN) of the pancreas undergoing surgical resection, with a focus on 10 patients who had aggressive features defined by local invasion of adjacent organs or development of metastases.  While this is one of the larger series of SPN, numerous other studies have performed similar analysis, limiting the impact and novelty of the current work.  Furthermore, changes outlined below would strengthen the paper.

1) What was the length of follow up for the series?

2) Figure 1 is missing

3) Long term outcomes are likely to be significantly different for locally invasion SPN and metastatic, therefore including both in this series may confound results.  I understand there are limited numbers of patients in each category, but I would be interested in seeing these two groups separated for analysis. 

4) Was locally invasion into an adjacent organ confirmed histologically or grossly during surgery?

5) Five disease specific deaths were reported in the table, but 7 in the text.

6) Neutrophil to lymphocyte ratio can be associated with aggressiveness as well (Yang et al, J Surg Onc 2019, PMID 31041808) and should be included in the current study

7) The discussion could be enhanced by including the following points:

-Addressing the limitation of selection bias since only surgical patients were included.

-Referencing more recent translational studies examining the biology of SPNs that may predict aggressiveness (MicroRNA PMID 32232006, whole exome sequencing PMID 28054945).

- Comparing and contrasting the current study findings to the previously reported literature

- Brief discussion as to how this information might influence clinical care/decision making

7) Since the PET classification is highlighted throughout the paper, it would be useful to see representative PET images of each of the types.

8) Extensive editing for grammar, syntax and diction is necessary

Author Response

Author response to the reviewers' comments

Thanks for your response and the reviewers' comments on our manuscript. We tried to faithfully revise our manuscript according to each question and concern raised by the reviewers, hoping that you will consider this manuscript for publication in Journal of Cancers

<Reviewer #1>

This is a series of 98 patients with solid pseudopapillary neoplasm (SPN) of the pancreas undergoing surgical resection, with a focus on 10 patients who had aggressive features defined by local invasion of adjacent organs or development of metastases. While this is one of the larger series of SPN, numerous other studies have performed similar analysis, limiting the impact and novelty of the current work. Furthermore, changes outlined below would strengthen the paper.

Comment 1 : What was the length of follow up for the series?

Response 1 : As commented by reviewr, we added the follow up period of each cases in table 4.

Revision 1 :

(Page 5,6)

Table 4. Profile of the patients with aggressive solid pseudopapillary tumors.

No.

Age

(years)

Sex

Tumor size (cm)

Symptom

Tumor location

Type of operation

Lymphovascular invasion

Type of aggressive behavior

Treatment for Recurrence

Follow-up period

(status)

1

81

F

12

Abdominal mass

Distal

Distal pancreatectomy

None

Liver metastasis after 5 months

Conservative treatment

21 months

(Death)

2

12

F

10.10

Abdominal mass

Distal

Distal pancreatectomy

None

Liver metastasis after 10 months

Chemotherapy

115 months

(Death)

3

46

F

2.00

Incidental

Distal

Distal pancreatectomy

None

Peritoneal metastasis after 37 months

Chemotherapy

53 months

(Death)

4

67

M

2.5

Abdominal pain

Distal

Distal pancreatectomy

None

Liver metastasis after 43 months

Chemotherapy

70 months

(Survival)

5

45

F

11

Abdominal pain

Distal

Biopsy

None

Liver metastasis after 2 months

Chemotherapy

9 months

(Death)

6

46

F

4.3

Abdominal pain

Proximal

PPPD + metastasectomy

None

Liver metastasis on diagnosis; peritoneal metastasis after 41 months

Chemotherapy

58 months

(Survival)

7

68

F

5

Abdominal pain

Proximal

PPPD + right hemicolectomy, PV resection

None

Invading hepatic flexure, portal vein

No recurrence

0 months

(Death)

8

52

M

11

Abdominal pain

Proximal

PPPD + right hemicolectomy

None

Invading hepatic flexure

No recurrence

67 months

(Survival)

9

10

F

6

Abdominal pain

Distal

Distal pancreatectomy + transverse colectomy

None

Invading transverse colon

No recurrence

49 months

(Survival)

10

27

M

9.4

Abdominal mass

Proximal

PPPD + PV resection

None

Invading portal vein

No recurrence

39 months

(Survival)

PPPD, pylorus-preserving pancreaticoduodenectomy;

Comment 2 : Figure 1 is missing

Response 2 : Thanks for letting me know. We corrected this and inserted the figure in the right place.

Revision 2 :

(Page 5)

Figure 1. Disease-specific Survival plot for Aggressive versus Non-aggressive SPN        

SPN, Solid pseudopapillary neoplasm;

Comment 3 : Long term outcomes are likely to be significantly different for locally invasion SPN and metastatic, therefore including both in this series may confound results. I understand there are limited numbers of patients in each category, but I would be interested in seeing these two groups separated for analysis.

Response 3 : As commented by reviewr, we classifed the aggressive SPN into Metastatic SPN and Locally invasion SPN groups and analyzed them for comparison. As a result of the analysis, there was no statistically significant difference in clinicopathological characteristics and disease-specific survival of the two groups. This information added as supplemental data.

(Supplemental data)

Table 1. Characteristics of patients with metastatic and Locally invasion solid pseudopapillary tumor.

Variables

Metastatic SPN (N=6)

Locally invasion SPN (N=4)

p-value

Age

≥40*

5 (83.3%)

2 (50%)

0.500

19-39

0

1 (25%)

>0.999

≤18

1 (16.7%)

1 (25%)

>0.999

Sex (F/M)

5 / 1

2 / 2

0.500

BMI

21.2 ± 4.45

22.5 ± 5.74

0.710

Symptomatic (yes/no)

5 / 1

4 / 0

>0.999

Location (proximal/distal)

1 / 5

3 / 1

0.190

Tumor size

≥10 cm

3 (50%)

1 (25%)

0.571

≥5 cm

3 (50%)

3 (75%)

0.571

≥2 cm

5 (83.3%)

4 (100%)

>0.999

PET configuration*

Type III

3 (60%)

4 (100%)

>0.999

Non-type III

2 (40%)

0

>0.999

Complication (yes/no)

1 / 5

3 / 1

0.190

Microscopic pathology

Margin

All negative

All negative

Capsular invasion

1 (16.7%)

1 (25%)

>0.999

Lymphovascular invasion

2 (33.3%)

1 (25%)

>0.999

Perineural invasion

2 (33.3%)

1 (25%)

>0.999

Ki-67

4.63 ± 5.31

1.00 ± 0.0

0.266

Values are n (%), mean ± standard deviation

PET, positron emission tomography; BMI, body mass index; WBC, white blood cell; CI, confidence interval;

Figure 1. Disease-specific Survival plot for Aggressive versus Non-aggressive SPN                     SPN, Solid pseudopapillary neoplasm;

Comment 4 : Was locally invasion into an adjacent organ confirmed histologically or grossly during surgery?

Response 4 : Local invasion into adjacent organ of the aggressive SPNs was histologically confirmed at all. We added this information in results.

Revision 4 :

(Page 5)

Ten patients (7 women, 3 men) had aggressive SPN; of these, 6 had systemic recurrences, while the remaining 4 had locally infiltrative neoplasms affecting adjacent organs confirmed histologically.

Comment 5 : Five disease specific deaths were reported in the table, but 7 in the text.

Response 5 : Thank you for letting me know. As mentioned in text, seven disease specific deaths are correct. Corrected the errors listed in the table.

Revision 5 :

(Page 4)

Table 3. Surgical outcomes of all 98 patients.

Variables

No. of patients (%)

Disease-specific deaths

7 (7.1%)

Comment 6 : Neutrophil to lymphocyte ratio can be associated with aggressiveness as well (Yang et al, J Surg Onc 2019, PMID 31041808) and should be included in the current study

Response 6 : As commented by reviewr, we added the suggested paper as recent paper in introduction.

Revision 6 :

(Page 2)

Several studies have attempted to identify factors that could predict the aggressive behavior of SPNs, including neutrophil-to-lymphocyte ratio [1].

  1. Yang, F.A.-O.; Bao, Y.; Zhou, Z.; Jin, C.; Fu, D. Preoperative neutrophil-to-lymphocyte ratio predicts malignancy and recurrence-free survival of solid pseudopapillary tumor of the pancreas.

Comment 7 : The discussion could be enhanced by including the following points:

7.1. Addressing the limitation of selection bias since only surgical patients were included.

7.2. Referencing more recent translational studies examining the biology of SPNs that may predict aggressiveness (MicroRNA PMID 32232006, whole exome sequencing PMID 28054945).

7.3. Comparing and contrasting the current study findings to the previously reported literature

7.4. Brief discussion as to how this information might influence clinical care/decision making

Response 7 : We added everything the reviewers mentioned to the discussion.

Revision 7 :

(Page 8)

7.1. This study has limitations related to its retrospective design and the small size of da-ta. Moreover, there was a selection bias since we only analyzed patients who underwent surgery.

7.2. Meng et al. conducted whole exome sequencing in 9 patients with SPN and found that the CTNNB1 mutation potentially collaborated with other gene variations  [2]. Shmuel et al. reported that a panel of 6 mi RNAs, including miR-184, miR-10a, miR-887, miR-217, miR200c, and miR-375, were significantly expressed in metastatic SPNs. These specific miRNAs have potential as predictive markers of aggressive behavior of SPN [3] .

  1. Guo, M.; Luo, G.; Jin, K.; Long, J.; Cheng, H.; Lu, Y.; Wang, Z.; Yang, C.; Xu, J.; Ni, Q. et al. Somatic genetic variation in solid pseudopapillary tumor of the pancreas by whole exome sequencing. Lid - 10.3390/ijms18010081 [doi] lid - 81.
  2. Cohen, S.J.; Papoulas, M.; Graubardt, N.; Ovdat, E.; Loewenstein, S.; Kania-Almog, J.; Pasmanik-Chor, M.; Brazowski, E.; Cagnano, E.; Nachmany, I. et al. Micro-rna expression patterns predict metastatic spread in solid pseudopapillary neoplasms of the pancreas. Front Oncol 2020, 10, 328-328.

7.3. Similar to our findings, in a multicenter analysis in Korea, large tumors (>8 cm) were reported to be predictive of recurrence [Exp(β)=7.385, p=0.018] [4]. Several studies have reported the importance of PET/CT as a predictor of aggressive behavior in in SPNs [5,6]. Aisheng et al. reported that CT or MRI demonstrated morphological features of SPN and FDG PET/CT reflected the histopathological composition of the tumors [5]. They explained that FDG uptake by SPN may be related to tumor cellularity, proliferative index, or histological malignancy [5].

Contrary to our results, some studies reported that young age and male sex were associated with SPN recurrence [7,8]. Sabine et al. reported that younger children had a high risk of recurrence (p=0.03) [7]. Matthew et al. reported that male patients have an atypically aggressive biology of SPN Males had approximately twice the rates of metastases and invasive malignancy and a threefold higher death rate than that of females (p=0.036, p=0.003, p=0.002, respectively) [8]. In our study, there was no difference in the incidence of aggressive SPN in younger individuals under 18 years of age (p=0.664), and was identified as a risk factor for individuals who were 40 years of age or older (Exp(β)=23.242, p=0.015). In addition, there was no statistically significant difference between males and females for aggressive SPN (p=0.173).

  1. Kang, C.M.; Choi Sh Fau - Kim, S.C.; Kim Sc Fau - Lee, W.J.; Lee Wj Fau - Choi, D.W.; Choi Dw Fau - Kim, S.W.; Kim, S.W. Predicting recurrence of pancreatic solid pseudopapillary tumors after surgical resection: A multicenter analysis in korea.
  2. Dong, A.; Wang Y Fau - Dong, H.; Dong H Fau - Zhang, J.; Zhang J Fau - Cheng, C.; Cheng C Fau - Zuo, C.; Zuo, C. Fdg pet/ct findings of solid pseudopapillary tumor of the pancreas with ct and mri correlation.
  3. Kim, Y.I.; Kim, S.K.; Paeng, J.C.; Lee, H.Y. Comparison of f-18-fdg pet/ct findings between pancreatic solid pseudopapillary tumor and pancreatic ductal adenocarcinoma.
  4. Irtan, S.; Galmiche-Rolland, L.; Elie, C.; Orbach, D.; Sauvanet, A.; Elias, D.; Guérin, F.; Coze, C.; Faure-Conter, C.; Becmeur, F. et al. Recurrence of solid pseudopapillary neoplasms of the pancreas: Results of a nationwide study of risk factors and treatment modalities.
  5. Lin, M.Y.; Stabile, B.E. Solid pseudopapillary neoplasm of the pancreas: A rare and atypically aggressive disease among male patients.

7.4. The results of this study presented clinical predictors, including PET/CT, to provide the basis for implementing PET/CT when treating patients with SPN in clinical practice. In patients with symptoms at diagnosis, large tumors (≥10 cm), and old age (≥40 years), PET/CT can be performed to check for aggressive behavior. The type III uptake pattern of PET/CT can be considered as an indicator for active surgery and postoperative chemotherapy. To the contrary, patients with type IV and V, mostly containing defective background with minimal or null FDG uptake, could be carefully followed without surgery according to the patients’ individual conditions, such as co-morbidity, refusal of surgery, and active personal busy schedule [9]. These SPNs may show very indolent biological behavior or total necrosis [9]. Therefore, the results of this study are expected to help SPN treatment decisions.

  1. Park, M.; Hwang, H.K.; Yun, M.; Lee, W.J.; Kim, H.; Kang, C.M. Metabolic characteristics of solid pseudopapillary neoplasms of the pancreas: Their relationships with high intensity (18)f-fdg pet images. Oncotarget 2018, 9, 12009-12019.

Comment 8 : Since the PET classification is highlighted throughout the paper, it would be useful to see representative PET images of each of the types.

Response 8 : As commented by the reviewer, PET image is very important in this study. We added the representative PET/CT images of each type in table 2.

Revision 8:

(Page 3,4)

Table 2. Clinical pattern of preoperative PET/CT in all 86 patients.

PET/CT scan uptake classification1

Scheme

Description

No. of patients (%)

Representative PET/CT image

Type I

Completely hot uptake type, no defect area

24 (27.9%)

Type II

Focal defect area with almost type I background

18 (20.9)

Type III

Multiple, or geographic uptake with defective background

23 (26.7%)

Type IV

Focal uptake with mainly defective background

18 (20.9%)

Type V

No uptake, completely defective background

3 (3.5%)

Values are n (%), mean ± standard deviation.

PET/CT, positron emission tomography/computed tomography;

1PET/CT scan uptake classification was described in a previous report [10].

  1. Kang, C.M.; Cho, A.; Kim, H.; Chung, Y.E.; Hwang, H.K.; Choi, S.H.; Lee, W.J. Clinical correlations with (18)fdg pet scan patterns in solid pseudopapillary tumors of the pancreas: Still a surgical enigma? Pancreatology : official journal of the International Association of Pancreatology (IAP) ... [et al.] 2014, 14, 515-523.

Comment 9 : Extensive editing for grammar, syntax and diction is necessary

Response 9: As menstioned by reviwer, we edited for grammar

"Some pictures are not visible. Please see the attatchment"
